# Alleviating the Sample Selection Bias in Few-shot Learning by Removing Projection to the Centroid

**Jing Xu[1,3], Xu Luo[2], Xinglin Pan[2], Wenjie Pei[1], Yanan Li[4] , Zenglin Xu[1,3]**[*]
[1]Harbin Institute of Technology, Shenzhen
[2]University of Electronic Science and Technology of China
[3]Pengcheng Laboratory  [4]Zhejiang Lab
`jingxu.may@gmail.com, frank.luox@outlook.com, kp600168@gmail.com`
`wenjiecoder@outlook.com, ynli.zju@gmail.com, zenglin@gmail.com`

## Abstract

Few-shot learning (FSL) targets at generalization of vision models towards unseen tasks without sufficient annotations. Despite the emergence of a number of few-shot learning methods, the sample selection bias problem, i.e., the sensitivity to the limited amount of support data, has not been well understood. In this paper, we find that this problem usually occurs when the positions of support samples are in the vicinity of *task centroid*—the mean of all class centroids in the task. This motivates us to propose an extremely simple feature transformation to alleviate this problem, dubbed Task Centroid Projection Removing (TCPR). TCPR is applied directly to all image features in a given task, aiming at removing the dimension of features along the direction of the task centroid. While the exact task centoid cannot be accurately obtained from limited data, we estimate it using base features that are each similar to one of the support features. Our method effectively prevents features from being too close to the task centroid. Extensive experiments over ten datasets from different domains show that TCPR can reliably improve classification accuracy across various feature extractors, training algorithms and datasets. The code has been made available at `https://github.com/KikimorMay/FSL-TCBR`.

## 1 Introduction

Deep networks have made great progress in image classification [16, 14, 47, 31]. However, large-scale annotated datasets are costly and even infeasible to obtain in many real-world applications. To explore the possibility of vision models to learn visual concept quickly, few-shot learning (FSL) [10, 41] has raised attention recently. In this problem, a vision model, trained on a base dataset with labeled data, needs to recognize novel classes of data (query set) given only a few labeled images (support set).

Typical few-shot learning methods usually first learn a good feature extractor from the base dataset. Then in each test-time task, the extracted features of all images in the support set are used to construct a linear classifier to recognize category of images from the query set [40, 8]. While a good feature extractor may help cluster unseen data, the task distribution shift between training and testing [25] still makes it hard to estimate novel class distribution using a small number of samples from the support set. Thus, the performance is strongly correlated with the sample quality of the support data. If the support data are not typical enough to represent the whole class, or if there exists outlier data that is not representative, the constructed classifier will be biased, which is called the *sample selection bias* problem [6]. In this paper, we delve into this problem in the context of few-shot learning, and find that the *task centroid*, defined as the mean of all class centroids in a task, plays

---

[*]Corresponding author

36th Conference on Neural Information Processing Systems (NeurIPS 2022).

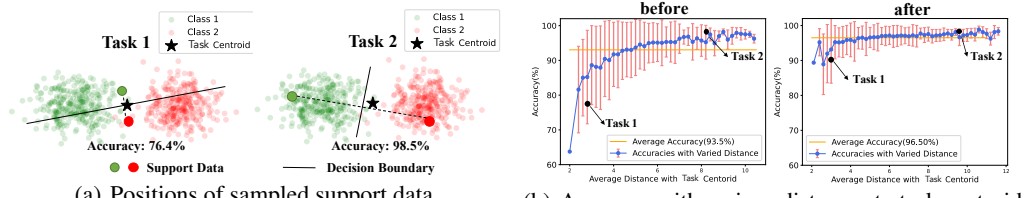

(a) Positions of sampled support data     (b) Accuracy with various distances to task centroids

Figure 1: 1(a) The decision boundary is extremely sensitive to the position of support features in FSL, resulting in high performance fluctuations, especially when they are in the vicinity of the task centroid. 1(b) The correlation between the classification accuracy and the average distance of all support data to the task centroid. The accuracy gets lower and the variance gets larger when the support samples are closer to the task centroid. After applying the proposed transformation, the bias is alleviated.

an important role. Intuitively speaking, the direction of the task centroid can be regarded as the commonality (or the shared pattern) among all classes in that task. Thus a support feature close to the task centroid contains more ambiguous information that may confuse the classifier. We demonstrate this phenomenon in Figure 1(a), where we consider the 1-shot case—each class contains one support sample. As seen, when these samples are close to the task centroid, the constructed classifier is very likely to be biased—a small perturbation to the support sample will lead to a large rotation of the classification boundary.

To verify our intuition, we randomly choose two classes "Lion" and "Dalmatian" from the test set of *mini*Imagenet, and randomly sample 10000 binary 1-shot classification tasks using these two classes. In Figure 1(b), we show how accuracy varies with the distance of support data to the task centroid. As we expected, a small distance of support data to the task centroid generally results in low average accuracy (plotted in blue) while producing high variance (plotted in red).

To alleviate this problem, we propose a simple transformation on features that removes the projection on the direction of the task centroid, named Task Centroid Projection Removing(TCPR). Since the ground-truth task centroid is not available and a small amount of support data is inadequate for obtaining a good approximation, we seek the assistance of the base set based on the observation: features from a novel dataset tend to cluster around a direction in the feature space [39], and a small number of base features close to that direction may be strongly correlated with these features. Inspired by this, we estimate the direction using the most similar top $k$ base samples to the support data. Since our proposed transformation removes the projection to the centroid, the impact of the commonality of classes is mitigated and the feature space becomes more discriminative for these classes. After transformation, the problem is relieved as shown in Figure 1(b). The transformation helps the classifier capture the object of interest from the misleading shared patterns (see Figure 6).

Our method is generic, flexible, and can be used agnostic to the pre-training based FSL methods. Empirical experiments show that the transformation function can consistently and largely boost the classification performance on ten datasets with various gaps between base and novel classes, demonstrating the importance of reducing the widely existing sample selection bias. Our contributions are summarized as follows: 1) To the best of our knowledge, we are the first to unravel the connection of the sample selection bias probelm with the task centroid in few-shot learning which has been ignored so far; 2) To reduce the sample selection bias, we propose a simple transformation to remove projection to an approximated task centroid estimated with the assistance of related neighbors in the base set, and 3) Comprehensive experiments show the effectiveness of the proposed transformation.

## 2 Related Work

Most recent few-shot learning approaches can be categorized into three groups: optimization-based, generation-based and metric-based methods. Optimization-based methods aim to learn models with good initialization so that the model can quickly adapt to novel set with a limited number of gradient descent on a few labeled examples, including MAML [10, 11], Reptile[30], LEO[36],Meta-B[5], MeTAL[2], $\mathcal{S}/\mathcal{T}$[22], etc. Generation-based methods deal with data deficiency problem by generating

pseudo support samples based on extra prior knowledge [45]. Some work utilize the transferable intra-class deformations in base set like SGM [12], VFD [48]and DC [49]. Other methods use high level semantic embedding of classes as extra prior information to generate novel samples, i.e., class-level attributes [50], word2vec [28, 52] or unlabeled query data [21]. Metric-based methods attempt to encode images into a well-shaped feature embedding space and perform classification with distance-based classifiers, e.g., MatchingNet[41], ProtoNet [37] and Relation Network[38]. Some studies reveal that the standard transfer learning paradigm can achieve surprisingly competitive performance in few-shot learning [4, 8, 7, 43]. In this paradigm, the models are trained on base set to learn high-quality feature extractors and only the classifiers are learned with novel samples. Based on this observation, several work consider improving the generalization ability of learned image representations on the base set with some auxiliary pretext tasks. Inspired by recently proposed self-supervised learning methods [13, 3], some FSL methods propose to use the contrastive learning with some regularization techniques to learn high-quality feature representation, e.g., S2M2 [27], Inv-Equ[34], VLCL [23], PAL [26], ArL [52] and POODLE [17]. Those methods demonstrate the significance of powerful feature representations in FSL.

However, the generalization ability will be damaged when there exists a large distribution shift between base and novel classes, leading to bias in the estimation of feature distributions of novel classes [39]. Luo et al. [24] manifest that image background is one harmful knowledge that prevent models from accurately capturing feature distributions of novel classes. Luo et al. [25] reveal that channels of features may have different importance in different tasks, and the estimation bias of channel importance reduces the discrimination between novel classes. Some simple transformation functions reshape the skewed distribution of novel features to improve the classification performance. The $l_2$ normalization can remove the negative effect caused by the magnitude of features [4, 20, 40]. Based on that, SEN [29] with $l_2$ normalization transforms features onto the hypersphere while keeping features repel from all other prototypes. SimpleShot [44] subtracts the mean of whole training set features before $l_2$ normalization. However, when there exists a large distribution shift between base and novel set, the performance of SimpleShot drops quickly since the mean of all training set features is not always a good information for the various class distributions of novel set. ZN [9] discovers the existence of the hubness problem in FSL and uses ZN normalization to address it. For every component of an individual novel feature, the ZN transformation subtracts the mean of all components and then divides the standard deviation of all components. DCM [39] calibrates the distribution of novel samples to approach zero centered mean and unit standard deviation using support data to improve the evaluation performance. Compared with ZN and DCM, the motivation of our TCPR is different. We focus on reducing the classification bias when the support samples in the vicinity of local centroid. Besides, both ZN and DCM transform novel features without the help of base set, which may lose some useful information. On the other hand, using a few samples in novel classes also results in a bias in task sampling. Some studies [1, 49, 39, 46] show that modern few-shot learning algorithms are extremely sensitive to the data used for adaptation. However, none of them analyzes the connection of the sampling bias to the effect of task centroid.

## 3 Methods

### 3.1 Preliminaries

In few-shot classification, we are given a base set $\mathcal{D}_b$ and a novel set $\mathcal{D}_n$ with disjiont classes $\mathcal{C}_b$ and $\mathcal{C}_n$, i.e., $\mathcal{C}_b \cap \mathcal{C}_n = \emptyset$. Training takes place on $\mathcal{D}_b$, and few-shot evaluation takes place on $\mathcal{D}_n$. Each FSL evaluation episode consists of a $N$-way $K$-shot task, which evaluates model's ability to learn a new classifier discriminating $N$ novel classes of $\mathcal{D}_n$ with only $K$ labeled data available in each class. The $N \times K$ labeled samples comprise the support set $\mathcal{S} = \{(x_i, y_i)\}_{i=0}^{N \times K}$. Based on the labeled support set $\mathcal{S}$ and the base dataset $\mathcal{D}_b$, our goal is to predict the category of all the unlabeled samples in the query set $\mathcal{Q} = \{(x_i, y_i)\}_{i=N \times K+1}^{N \times K + N \times q}$ correctly, where $q$ denotes the number of images for each class in the query set. The query set is sampled from the same $N$ novel classes as $\mathcal{S}$ in each task.

Our work is based on the popular pre-training pipeline. The sufficient labeled base samples are used for pre-training. For novel classes, we fix the parameters $\theta$ in feature extractor $f_\theta$ unchanged. We learn a new classifier $f_{\mathbf{W}}$ using the limited samples in the support set $\mathcal{S}$ to predict labels for query samples $\mathcal{Q}$. Our approach is of feature-level and is independent of the pre-trained backbones.

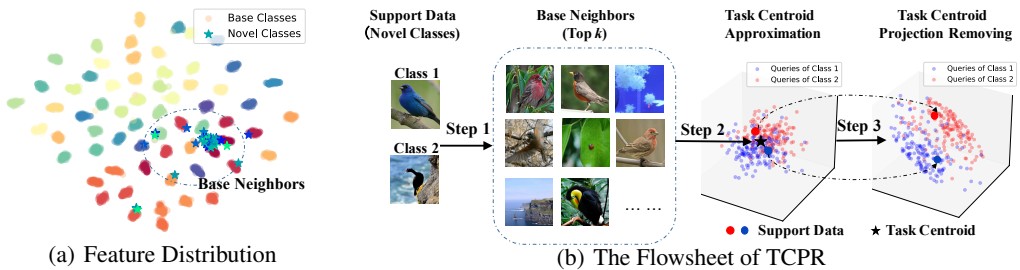

(a) Feature Distribution

(b) The Flowsheet of TCPR

Figure 2: 2(a) The feature distribution of base classes(*mini*Imagenet) and novel classes(CUB) show the novel classes skew to some direction and are more related to limited base samples. 2(b) The overall process of TCPR. Given novel support samples, our TCPR consists of three steps: 1) seeking the most related top $k$ samples in base set; 2) approximating the task centroid of the novel data; 3) removing the harmful effect of the direction of the task centroid. The visualization is conducted by randomly choosing two classes and projecting all data into a 3-dimensional subspace using PCA.

**Pre-training Stage:** Recently, some studies [40, 8] have demonstrated that the significance of powerful feature representations with a simple feature embedding network $f_\theta$ learned on base classes through a image classification proxy task has good transferability to novel samples. Furthermore, to strengthen the generalization of $f_\theta$, the self-supervised learning with other proxy tasks, e.g., colorization and rotation, is used as auxiliary loss in FSL. To show our method is agnostic to the feature extractor, we implement the proposed transformation on different algorithms, including Baseline++ [4]( which trains $f_\theta$ with vanilla cross-entropy loss); S2M2 [27](which uses mixup regularization and an auxiliary task to predict the rotation angles); Inv-Equ [34] (which uses sixteen geometric transformations for self-supervised learning and knowledge distillation techniques).

**Evaluation stage:** For each $N$-way $K$-shot task, we train a new linear classifier $f_\mathbf{W}$ using the support set $\mathcal{S}$ with $N \times K$ limited labeled samples. Let $\mathbf{W} = [w_1, w_2, ..., w_N] \in \mathbb{R}^{d \times N}$ denotes the learn-able parameters of the classifier, where $d$ is the dimension of the extracted features. We normalize the weight vector for each class $c$, where $||w_c|| = 1, c \in [1, 2, ..., N]$, to eliminate the effect of the magnitude of the feature vector[42]. The probability of a sample $x$ belonging to class $c$ can be formulated as:

$$P(y = c|x) = \frac{e^{\gamma(w_c)^T x}}{\sum_{i=1}^{N} e^{\gamma(w_i)^T x}}, \tag{1}$$

where $\gamma$ is a scaling hyper-parameter. After training on $\mathcal{S}$, the weights vectors $[w_1, w_2, ..., w_N]$ can be seen as prototypes of novel classes and used to predict the samples in the query set $\mathcal{Q}$. The classification result is based on the distance of the query image features to these learned prototypes.

Although the simple pre-training strategy has achieved promising performance with well-designed feature extractors, the ambiguity in classification caused by the few samples in the novel classes is still unsolved. Besides, the ambiguity is further aggravated in the vicinity of the task centroid.

### 3.2 Task Centroid Projection Removing

In this work, we propose a simple transformation to remove the sample bias exacerbated by the task centroid, named Task Centroid Bias Removing (TCPR). The overview of the proposed TCPR is shown in Figure 2(b), which consists of three steps: 1) given support data $\mathcal{S}$ from novel classes, we find the most related top $k$ samples in the base set; 2) using the top $k$ neighbors, we approximate the task centroid vector with the help of the statistics of the base set, and 3) we alleviate the harmful effect of the approximated centroid by removing the component of novel features along the direction.

**Step 1: Seeking the base neighbors.** Our work is based on the phenomenon that the domain shift between base and novel sets causes the skewness of novel class distribution to some specific directions [39]. As shown in Figure 2(a), we visualize the distributions of the base dataset, i.e., *mini*Imagenet (which is a coarse-grained dataset with a wide range of object categories) and the

novel dataset, i.e., CUB (which is a fine-grained dataset that only contains various bird categories), showing that the novel classes are close to only some specific base classes and skew to the direction. Inspired by the observation, we try to approximate the task centroid by looking back upon similar base samples. In the $N$-way $K$-shot task, given novel support normalized features $x_i^n \in \mathcal{S}, ||x_i^n|| = 1, i \in [1, 2, ..., N \times K]$, extracted by the pre-trained feature extractor $f_\theta$, we calculate the mean of $\mathcal{S}$ by averaging the extracted features of all support samples, that is,

$$\bar{x}^n = \frac{1}{N \times K} \sum_{i=1}^{N \times K} x_i^n. \tag{2}$$

Using the mean of the support features, we seek the top $k$ base samples with the closest angular similarity (evaluated by the cosine distance) between the $\bar{x}^n$ and $x_i^b \in \mathcal{D}_{base}, i \in [1, 2, ..., N_{base}]$:

$$\mathcal{D}_{cosine} = \left\{ cos(\bar{x}^n, x_i^b) | i \in \{1, 2, ..., N_{base}\} \right\} \tag{3}$$
$$\mathcal{D}_{topK} = \left\{ i | cos(\bar{x}^n, x_i^b) \in topK(\mathcal{D}_{cosine}) \right\},$$

where the notation $cos(\cdot, \cdot)$ denotes the cosine distance between two vectors and $topK(\cdot)$ is an operator to select the largest $k$ elements in the distance set $\mathcal{D}_{cosine}$[2]. The set $\mathcal{D}_{topK}$ contains the most related $k$ samples in base classes.

**Step 2: Approximating the task centroid.** We approximate the task centroid by a weighted aggregation of $x_i^b, i \in \mathcal{D}_{topK}, i \in \{1, 2, ..., k\}$ based on the cosine similarity in $\mathcal{D}_{cosine}$. The weight $w_i$ of each $x_i^b$ can be constructed as:

$$w_i = \frac{cos(\bar{x}^n, x_i^b)^p}{\sum_{i \in \mathcal{D}_{topK}} cos(\bar{x}^n, x_i^b)^p}, \tag{4}$$

where $p$ is the hyper-parameter controlling the relative weights of the base samples. The approximated task centroid $\mathbf{c}_{task}$ is formulated as:

$$\mathbf{c}_{task} = L_2 \left( \sum_{x_i \in \mathcal{D}_{topK}} w_i x_i^b \right), \tag{5}$$

where the notation $L_2(\cdot)$ denotes the $l_2$ normalization. The approximation of task centroid by the base neighbors, $\mathbf{c}_{task}$, is more accurate and stable than the means of support set, $\bar{x}^n$. With only a few examples, $\bar{x}^n$ is easy to overfit on the support data and far from mirroring the ground truth task centroid of the novel classes. Since the novel classes skew in a certain direction toward relevant limited base samples, extrapolating the task centroid from the top $k$ similar samples in base set can be seen as a more precise alternative. We can change the hyper-parameter $k$ to cover the distribution of novel classes with different domain gaps.

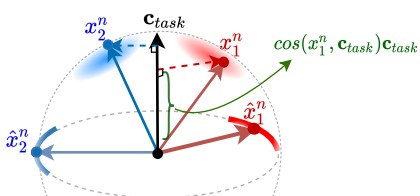

Figure 3: Removing the projection along the $\mathbf{c}_{task}$. The distribution gap between two classes 1(colored in red) and class 2(colored in blue) becomes larger.

**Step 3: Removing the projection to the task centroid.** To reduce the sampling bias caused by the support samples in the vicinity of the task centroid, we propose to eliminate the effect of the direction of the task centroid by pushing novel features away from it. For all the novel samples in both support and query set, $x^n \in \mathcal{S} \cup \mathcal{Q}$, we use a simple transformation which extracts the projection to the approximated task centroid $\mathbf{c}_{task}$:

$$\hat{x}^n = L_2(x^n - cos(x^n, \mathbf{c}_{task})\mathbf{c}_{task}). \tag{6}$$

After removing the component of features along the direction of $\mathbf{c}_{task}$, the distribution of novel classes is pushed away from the approximated centroid, shown in Figure 3.

Since the direction of task centroid can be seen as the commonality of classes of each task, it contains the non-discrimintaive information which inhibits the classification results. TCPR projects out the harmful direction and constructs a better latent space.

---

[2]The complexity of calculating the similarity matrix is $\mathcal{O}(N_{base} \times d)$, and selecting the $topK$ feature is $\mathcal{O}(N_{base} \times log(k))$. For empirical verification, the increase of latency is small (+11 ms) compared to the original time of training the classifier (298ms) for one 5-way 1-shot meta-testing task. The latency is averaged over tasks.

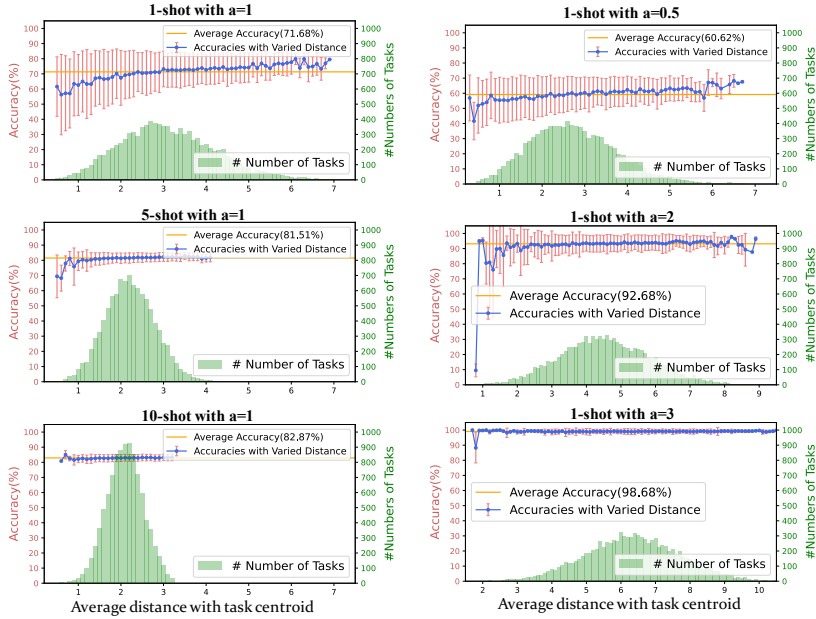

(a) Sampling bias with different number of shots

(b) Sampling bias with different value of $a$

Figure 4: Classification accuracy vs the average distance between two class prototypes to the centroid.

## 3.3 The Connection of Task Centroid to Ambiguity in FSL

In this subsection, we analyze the classification ambiguity aggravated by the task centroid, and further reveal that it is a naturally-occurring phenomenon in FSL since the limited samples result in a poor estimation of the class prototype, and the small distribution margin among novel classes makes the ambiguity more severe. For a better understanding, we design a toy simulation of a binary classification task. Specifically, let $\mathcal{D}_1 = \mathcal{N}(\boldsymbol{\mu}_1, \boldsymbol{\Sigma}_1)$, $\mathcal{D}_2 = \mathcal{N}(\boldsymbol{\mu}_2, \boldsymbol{\Sigma}_2)$ be two Gaussian distributions, where $\boldsymbol{\mu}_1 = [-a, 0]$, $\boldsymbol{\mu}_2 = [a, 0]$ and $\boldsymbol{\Sigma}_1 = \boldsymbol{\Sigma}_2 = \mathbf{I}$. The task is simply to discriminate between samples from the two classes. The task centroid of the two classes is the origin point $(0, 0)$. We use Nearest-Centroid Classifier [37] without any regularization to learn the decision boundaries, which first averages points from each class in the support set to form class prototypes and assigns query points to the class of the nearest prototypes. We randomly sample $10,000$ tasks for each simulation experiment. The connection between classification accuracy and the average distance from two class prototypes to the task centroid is shown in Figure 4. With the decrease of the distance, the average of accuracy results gets lower, while the variance of accuracy results gets larger, as shown in the 1-shot task with $a = 1$.

Intuitively, the sampling bias in the vicinity of task centroid ties with the number of shots in the FSL task. With the increase of the shots, the variance of the class prototypes will diminish. In $K$-shot task, the covariance matrix of class prototype is $\boldsymbol{\Sigma} = \mathbf{I}/K^2$. When the number of shots is large enough, the class prototypes can estimate the class centroid accurately with a low variance and the sampling bias is alleviated. This phenomenon is clearly shown in Figure 4(a) when $K = 10$. The poor estimation of the class prototypes caused by limited labeled data is one of the reasons why the sampling bias decreases the performance of FSL.

On the other hand, the bias correlates with the discriminability of the two classes' distribution. We vary the value of $a$ from $a = 0.5$ to $a = 3$ with a wide range to simulate distribution with different discriminability in 1-shot tasks. The two classes are more well-separated with larger $a$. As shown in Figure 4(b), the bias problem begins to emerge with $a = 0.5$, when there exists a large overlap between two classes' distribution. With the increase of $a$, the bias problem is more severe at first and is relieved later. It almost disappears when $a = 3$, where the inter-class distance is much larger than the intra-class variance. The feature distribution which is suffering from the sampling bias is highly corresponding to the few-shot learning classification problem. Recent studies have shown that the

Table 1: The 5-way 1/5-shot classification accuracies on *mini*ImageNet. The * denotes we randomly crops 9 patches for testing proposed in DeepEMD [51].

| Methods | Backbones | 5-way 1-shot | 5-way 5-shot |
|---|---|---|---|
| MAML [10] | ResNet-18 | $49.61 \pm 0.92$ | $65.72 \pm 0.77$ |
| MatchingNet [41] | ResNet-18 | $52.91 \pm 0.88$ | $68.88 \pm 0.69$ |
| ProtoNet [37] | ResNet-18 | $54.16 \pm 0.82$ | $73.68 \pm 0.65$ |
| DeepEMD [51] | ResNet-12 | $65.91 \pm 0.82$ | $82.41 \pm 0.56$ |
| DeepEMD* [51] | ResNet-12 | $68.77 \pm 0.29$ | $84.13 \pm 0.53$ |
| CSEI [19] | ResNet-12 | $67.59 \pm 0.83$ | $81.93 \pm 0.36$ |
| ArL [52] | ResNet-12 | $65.21 \pm 0.58$ | $80.41 \pm 0.49$ |
| MeTAL[2] | ResNet-12 | $66.61 \pm 0.28$ | $81.43 \pm 0.25$ |
| $\mathcal{S}/\mathcal{T}$[22] | ResNet-12 | $68.03 \pm 0.52$ | $82.53 \pm 0.47$ |
| POODLE [17] | ResNet-12 | $67.80$ | $83.50$ |
| PAL [26] | ResNet-12 | $69.37 \pm 0.64$ | $84.40 \pm 0.44$ |
| DC [49] | WRN-28 | $68.57 \pm 0.55$ | $82.88 \pm 0.42$ |
| PAL [26] | ResNet-12 | $69.37 \pm 0.64$ | $84.40 \pm 0.44$ |
| COSOC [24] | ResNet-12 | $69.28 \pm 0.49$ | $85.16 \pm 0.42$ |
| Baseline++ [4] | ResNet-18 | $54.23 \pm 0.46$ | $76.11 \pm 0.33$ |
| Baseline++ +TCPR | ResNet-18 | $\mathbf{58.67 \pm 0.41}$ | $\mathbf{77.31 \pm 0.32}$ |
| Baseline++ [4] | ResNet-12 | $61.67 \pm 0.42$ | $79.32 \pm 0.34$ |
| Baseline++ + TCPR | ResNet-12 | $\mathbf{64.92 \pm 0.40}$ | $\mathbf{80.87 \pm 0.33}$ |
| Inv-Equ[34] | ResNet-12 | $65.57 \pm 0.43$ | $84.43 \pm 0.28$ |
| Inv-Equ+TCPR | ResNet-12 | $\mathbf{68.92 \pm 0.44}$ | $\mathbf{85.32 \pm 0.25}$ |
| S2M2 [27] | WRN-28 | $64.63 \pm 0.43$ | $83.50 \pm 0.29$ |
| S2M2+TCPR | WRN-28 | $\mathbf{68.05 \pm 0.41}$ | $\mathbf{84.51 \pm 0.27}$ |
| S2M2* | WRN-28 | $67.19 \pm 0.45$ | $85.63 \pm 0.26$ |
| S2M2*+TCPR | WRN-28 | $\mathbf{70.67 \pm 0.43}$ | $\mathbf{86.61 \pm 0.26}$ |

high-quality image representations can be transferred from base to novel set, but there does not exist enough margin among novel classes due to the domain gaps. This leads to another factor causing the classification ambiguity aggravated by the task centroid. Similar phenomenon can be observed among the real data in higher dimensional space, as illustrated in Figure 1(b).

# 4   Experiments

**Datasets.** We evaluate our approach on four widely-used FSL datasets. The *mini*ImageNet[32] is a coarse-grained dataset with distinct classes of animals or objects. It is a subset of ILSVRC-12 [35], including 600 images of each class. Following the previous work [32], the categories are split into 64, 16, 20 classes for training, validation and evaluation respectively. *tiered*ImageNet is a much larger and more challenging dataset. It is made up of $779,165$ images from $608$ classes sampled from a hierarchical category structure. We adopt 351 classes as base categories, 97 as validation categories and 160 as novel categories as suggested in [33]. The **CUB** is a fine-grained dataset consisting of $11,788$ images from 200 bird classes. We spilt the dataset into 100, 50 and 50 categories as base, validation and test categories, following [4]. **Meta-dataset** is a significantly larger-scale dataset which comprised of multiple datasets of diverse data distributions, presenting more realistic tasks.

**Implementation Details.** We adopt the ResNet-18/12 [4, 34] and WRN [27] as the feature extractors $f_\theta$ for a fair comparison with published results. All the extracted features are normalized using the $l_2$ normalization. We use the cosine classifiers to learn the decision boundaries of the novel set. The $p$ in Equation 4 is set to be $0.5$. The number of related base neighbors are selected according to the degree of domain difference between base and novel set. We set a larger $k$ with a small task distribution shift, e.g., transferred from mini-train to mini-test, and a lower $k$ with a larger distribution shift, e.g., transferred from mini-train to Quick Draw. The details of the selection of $k$ can be found in Section 4.3 Hyper-parameter selection. All the ablation studies and analysis experiments are conducted with S2M2 [27]. We report the mean accuracy as well as the 95% confidence interval on 2000 randomly generated episodes.

Table 2: Results on *tiered*ImageNet.

| Methods | 5-way 1-shot | 5-way 5-shot |
|---|---|---|
| MAML [10] | $51.67 \pm 1.81$ | $70.93 \pm 0.08$ |
| LEO [36] | $66.33 \pm 0.05$ | $81.44 \pm 0.09$ |
| MetaOpt [18] | $65.81 \pm 0.74$ | $81.75 \pm 0.53$ |
| CSEI [19] | $72.57 \pm 0.95$ | $85.72 \pm 0.63$ |
| DeepEMD [51] | $71.00 \pm 0.32$ | $85.01 \pm 0.67$ |
| CAN [15] | $70.65 \pm 0.99$ | $84.08 \pm 0.68$ |
| Inv-Equ[34] | $72.21 \pm 0.90$ | $87.08 \pm 0.58$ |
| S2M2 | $74.87 \pm 0.43$ | $87.61 \pm 0.26$ |
| S2M2 + TCPR | $\mathbf{77.67 \pm 0.41}$ | $\mathbf{88.89 \pm 0.25}$ |

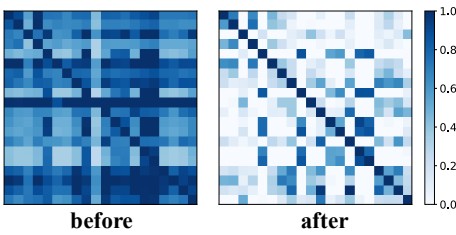

**before**       **after**

Figure 5: The cosine similarities among 20 novel class centers

Table 3: The 5-way 1-shot results on Meta-dataset.

| Test Set | mini-test | CUB | Fungi | Omini | Sign | QDraw | Flower | DTD |
|---|---|---|---|---|---|---|---|---|
| baseline | 64.63 | 47.75 | 42.36 | 77.28 | 53.50 | 51.60 | 70.33 | 50.47 |
| TCPR + $\bar{x}^n_{\mathcal{S} \cup \mathcal{Q}}$ | $69.52_{+4.89}$ | $53.83_{+6.08}$ | $46.28_{+3.92}$ | $80.88_{+3.60}$ | $56.65_{+3.15}$ | $57.31_{+5.71}$ | $75.37_{+5.04}$ | $54.38_{+3.91}$ |
| TCPR + $\bar{x}^n_{\mathcal{S}}$ | $66.88_{+2.12}$ | $50.48_{+2.73}$ | $43.15_{+0.82}$ | $77.89_{+0.77}$ | $54.22_{+0.72}$ | $54.60_{+3.00}$ | $70.33_{+-}$ | $51.49_{+1.03}$ |
| TCPR + $\mathbf{c}_{task}$ | $68.06_{+3.34}$ | $51.87_{+4.12}$ | $44.38_{+2.01}$ | $78.51_{+1.23}$ | $54.83_{+1.33}$ | $54.62_{+3.02}$ | $72.55_{+2.33}$ | $52.50_{+2.03}$ |
| SimpleShot [44] | $67.18_{+1.85}$ | $49.68_{+1.93}$ | $43.79_{+1.43}$ | $78.19_{+0.91}$ | $54.04_{+0.54}$ | $54.50_{+2.90}$ | $71.68_{+1.35}$ | $51.19_{+0.72}$ |
| ZN [9] | $67.05_{+2.42}$ | $48.15_{+0.40}$ | $43.24_{+0.88}$ | $78.80_{+1.52}$ | $53.92_{+0.42}$ | $52.86_{+1.20}$ | $70.21_{-0.21}$ | $52.20_{+1.73}$ |

## 4.1 Experimental Results

Note that any FSL method which follows the pre-training pipeline can be employed as the baseline in our method. To show our method is agnostic to feature extractors, we use the algorithms proposed in Baseline++ [4] , S2M2 [27], Inv-Equ [34] to obtain the embedding features as baseline models, without loss of generality. After applied TCPR transformation, we get $3\% - 4\%$ improvement on 1-shot task and nearly $1\%$ on 5-shot task. The improvements are larger in 1-shot task than 5-shot, since the sampling bias is more severe in 1-shot task, as shown in 3.3. Furthermore, we combine them to improve the generalization ability of brand-new images. Given original images, the strategy randomly samples 9 patches with different sizes and shapes, followed by resizing these patches to $84 \times 84$. We achieve remarkable $70.67\%$ and $86.61\%$ accuracies for 1/5-shot tasks showing that the most state-of-the-art FSL methods still suffer from the biased problem caused by task centroid and the problem can be alleviated by our method. The stable improvements validate the general applicability of TCPR for FSL. We also observe stable improvement on *tiered*Imagenet shown in Table 2.

## 4.2 Statistical Analysis

**The approximation of task centroids.** We approximate the task centroid of a novel task using four different statistic approximations, the mean of the support and query set $\bar{x}^n_{\mathcal{S} \cup Q}$, the mean of the support set $\bar{x}^n_{\mathcal{S}}$ and the weighted summation of base neighbors $\mathbf{c}_{task}$. We can observe from Table 3 that: (1) The $\bar{x}^n_{\mathcal{C}_{novel}}$ is the most accurate approximation of the task centroid, which is not available in inductive few-shot tasks and can be seen as the transductive setting. The improvements on all datasets by a large margin show the existence of the detrimental impact of task centroid, and it can be alleviated by our TCPR transformation; (2) Compared with the limited information in $\bar{x}^n_{\mathcal{S}}$, seeking the related samples in the base set as assistance, $\mathbf{c}_{task}$, can be a better estimation.

**The comparison with alternative normalization operations.** We also compare different transformations on Meta-dataset shown in Table 3 based on pre-trained models. The SimpleShot [44] subtracts the mean of whole training set features before $l_2$ normalization. The improvements are limited when the train-test gaps are large. The Z-score normalization [9] is applied to each feature vector independently, causing a loss of utilizing the task centroid information. Our proposed TCPR achieves better performance with the assistance of the approximation of task centroid, which makes it insensitive to the base-novel gap.

**The task centroid clusters different novel classes in the same direction and hurts the discriminability.** We visualize the cosine similarities among 20 novel class centers of *mini*Imagenet before and after the proposed transformation, as shown in Figure 5. It is clear that the centers of novel classes are scattered in the same direction (task centroid of the novel set), which forms clusters with

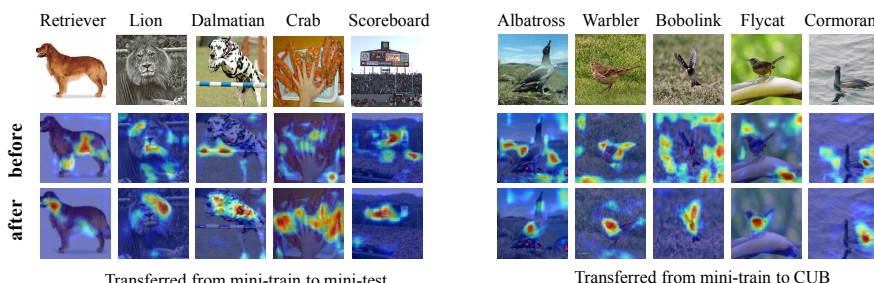

Transferred from mini-train to mini-test          Transferred from mini-train to CUB

Figure 6: The Grad-Cam class activation maps of query samples.

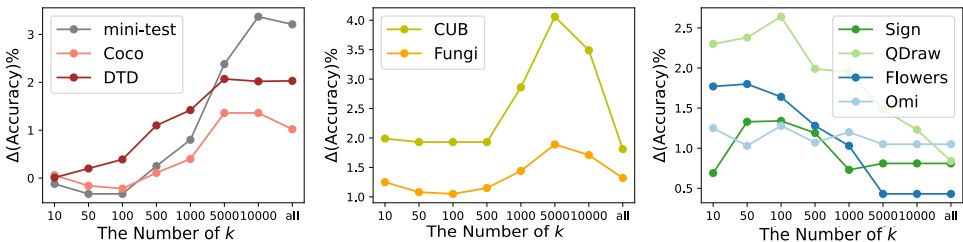

Figure 7: The effect of the number of base neighbors on different novel sets.

large cosine similarity before transformation. After the transformation removes the projection along the direction, samples of different classes in the feature space are more separable.

**The projection on the task centroid vector distracts the neural network from the main objects.** Figure 6 shows the Grad-Cam activation maps of some query samples with and without the TCPR transformation. We observe that our transformation by removing the projection of the task centroid helps the model adjust attention to the objects for classification with only a few support images. Before transformation, skewness of features makes the classifier confused by the shared structure and distracts the neural network from the main objects. After removing the projection on task centroid vector of task direction, it is easier for the classifier to capture the objects of interest.

### 4.3 Ablation Study

**Hyper-parameter selection.** Selection of the number $k$ of similar base neighbors plays an essential role in approximating the task centroid of novel classes. We show how hyperparameter $k$ influences the few-shot classification performance on different novel sets with varied domain differences in Figure 7. The optimal value of $k$ is different according to the difference of the base-novel domain. Evidently, datasets that are similar to the train split of *mini*Imagenet, like the test split of *mini*Imagenet and Coco, often result in a larger number of optimal $k$. Datasets with less domain gap require small $k$, like the QDraw and Omniglot. It is necessary to mention that the DTD is a texture dataset. So, it is hard to find the most related base samples, using the whole training set can be a better estimation. The results also manifest importance of a good approximation of task centroid since different approximations will cause significant change in accuracy .

**Performance with different number of ways and shots.** We show the results with different numbers of ways and shots before and after the simple transformation in Table 4. The transformation improvement is more significant when the number of shots is lower since the bias problem is more severe in the low-shot situation.

Table 4: The accuracy with a wide range of number of ways and shots.

| $N$-way $K$-shot | $N = 2$ | | | $N = 5$ | | | $N = 10$ | | |
|---|---|---|---|---|---|---|---|---|---|
| | $K = 1$ | $K = 5$ | $K = 10$ | $K = 1$ | $K = 5$ | $K = 10$ | $K = 1$ | $K = 5$ | $K = 10$ |
| before | 84.52 | 94.47 | 95.95 | 64.63 | 83.50 | 87.89 | 50.68 | 73.97 | 79.15 |
| after | 88.39 | 94.91 | 96.21 | 68.05 | 84.52 | 88.14 | 52.80 | 74.52 | 79.63 |

## 5 Conclusion and Future Work

In this work, we disclose a phenomenon that few-shot classification performance would be very sensitive to the position of support samples if they are in the vicinity of the task centroid, which leads a decrease of accuracy with a large variance in different tasks. A simple feature-level transformation, named TCPR, is proposed to address bias by removing the component of novel features to an approximated task centroid direction achieving consistent improvements on Meta-datasets. In this paper, we consider the sampling bias effect of task centroid. In the future, we will explore the influence of variance of data. Besides, a self-adaptive selection algorithm of the number of $k$ in base set is worth to explore to improve the generalization of TCPR. Furthermore, we plan to apply our method to dense prediction FSL tasks like object detection/segmentation in the near future.

## Acknowledgments and Disclosure of Funding

This paper was partially supported National Key Research and Development Program of China (No. 2018AAA0100204), and a key program of fundamental research from Shenzhen Science and Technology Innovation Commission (No. JCYJ20200109113403826), and the Major Key Project of PCL (No. PCL2021A06), Foundation of Zhejiang Province (No. LQ20F030007), and the Open Research Projects of Zhejiang Lab (NO.2022RC0AB04) and the National Natural Science Foundation of China (No. 62206256) .

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
