# OpenReview forum: "Alleviating the Sample Selection Bias in Few-shot Learning by Removing Projection to the Centroid"
_NeurIPS.cc/2022/Conference — NeurIPS 2022 Accept_

### Official Review · Reviewer_5bg4 · 2022-07-10

**Rating:** 4
**Confidence:** 5
**Soundness:** 3 good
**Presentation:** 3 good
**Contribution:** 2 fair

**Summary:**

This paper proposes a post-processing technique to improve the quality of prototypes used in few-shot learning (FSL). The main idea is to remove a bias, by first identifying a large set of neighbors from the base samples (training data used in designing the model) and performing a weighted averaging to identify the origin of decision space (my interpretation). Once the center is identified, the support and query samples are normalized according to the center (by first a weighted bias removal and then normalization). Experiments on various datasets show that the method helps in having a less biased decision space, hence improving the results.

**Questions:**

As for questions, I would like to ask

1- Can the method be used to address dense prediction FSL? if yes, how?

2- I am not convinced that the change of origin is the source of bias. Somehow I feel that the algorithm tries to use the base data to construct a better latent space. A natural question to ask is, can we consider the PCA space from the base samples as a way of denoising the decision space? That is, aside from the task-center, what will happen if the PCA space is also taken into account. Can you kindly comment on this?

**Limitations:**

It is not discussed but the method extensively uses training data towards its decisions so could potentially generate unfair and biased outputs

**Strengths And Weaknesses:**

Strengths
-------------

- the paper is written very clearly.
- focusing on classification, the authors did a comprehensive study and contrast their algorithm on a broad set of experiments, including meta-dataset.

Weaknesses
-------------

- despite the fact that the authors explained the motivation behind the work (closeness to a center), I have a lot of difficulties convincing myself that the so-called task centroid is the source of the problem. Overall, I think one can scale samples, and distances to the task-center cannot be the only reason behind improvements here. I do understand that the solution works on the hyper-sphere so do not crusade me but my point is, the relative closeness cannot be the sole reason behind the improvement obtained in this work.

- the method is quite hefty in compute as it requires identifying k-neighbors of support samples, where k in practice can be in the range 10,000. As a matter of fact, Fig.7 suggests the algorithm underperforms on mini-imagenet if k <100.

- the problem studied is solely classification.

---

> ### Author Response · Authors · 2022-08-02
> **Response to Reviewer 5bg4(1/2)**
>
> We appreciate the reviewer for the insightful comments. Below, please find our responses accordingly.
>
> ### **Q1:the method is quite hefty in computing as it requires identifying k-neighbors of support samples, where k in practice can be in the range 10,000. Fig.7 suggests the algorithm underperforms on mini-imagenet if k <100.**
>
> **A1**:The reviewer might think the TCPR will incur hefty extra time costs, which is not true.
> The complexity of calculating the similarity matrix is $\mathcal O(N_{base}\times d)$, and selecting the top-k feature is $\mathcal O(N_{base}\times log(k))$. Here $N_{base}$ denotes the number of features in base data and $d$ denotes the dimension of extracted features.  For empirical verification, we report latency (ms) of one meta-testing task for 5-way 1-shot classification on mini-Imagenet as below. For fair comparison, the latency is averaged over $2000$ tasks. As we see from this table, the increase of latency is really small ($11$ms) compared to the original time of training the classifier ($298$ms).
>
> To further reduce the computational cost, we can sample a subspace from the whole base data. By randomly sampling p(\%) features in each base class in the subspace, we show the computational latency and the classification performance below. The improvements are still significant when using only 1\% features in base classes on Meta-dataset with 1\% storage space (only 0.98M).
>
> | Selection Ratio| Latency| mini-test| CUB| Fungi| Omini| Sign| QDraw| Flower| DTD|
> |-|-|-|-|-|-|-|-|-|-|
> | baseline| 298| $64.63$| $47.75$| $42.36$| $77.28 $| $53.50$ | $51.60$| $70.33$| $50.47$
> |TCPR| + 11| $68.06$ | $51.87$ | $44.38$ | $78.51 $ | $54.83$ | $54.62$ | $72.55$ | $52.50$ |
> | TCPR(p=1\%)    | +4| $67.79$| $51.55$| $44.29$| $78.18$| $54.03$| $53.62$| $71.82$| $52.69$|
>
> We also want to emphasize that our TCPR still achieves better performance on most test sets except mini-test and coco if k<100, as shown in Figure 7.
>
> In summary, the computational cost of identifying k-neighbors of support samples is small and can be further reduced with a large margin of improvement over baseline methods.
>
> ### **Q2: I have a lot of difficulties convincing myself that the so-called task centroid is the source of the problem. Overall, I think one can scale samples, and distances to the task-center cannot be the only reason behind improvements here. I do understand that the solution works on the hyper-sphere so do not crusade me but my point is, the relative closeness cannot be the sole reason behind the improvement obtained in this work.**
>
> **A2**:  To better understand the role of the centroid, let's regard the task centroid as the commonality (or the shared pattern) among all novel classes. To achieve better performance in FSL,  one needs to make the model more discriminative while inhibiting the commonality of different classes. Hence, our method removes the commonality by simply removing the projection to the centroid.   As shown in Fig 5 where we visualize the pairwised similarity among 20 novel classes of the mini-test dataset,  the similarities get decreased after removing the projection to the centroid of the 20 classes, meaning that novel classes become more separated.  Furthermore, let's see Grad-Cam class attention map in Figure 6. After removing the projection to the centroid, the classifiers put more attention to the objects of interest in images, while neglecting some misleading shared pattern.
>
> In addition, we want to correct the reviewer that the scaling of data will result in equal scaling of the distance between every pair of samples, and does not affect the relative distance/similarity between pair of samples. For example, scaling the samples in Figure 1 by 10, 100, or even 1 million does not make the classification task easier, and the bias problem regarding the task centroid will stand as long as the overall data topology does not change. In contrast, our method adjusts the  relative closeness or topology  by removing the projection to the centroid, which cannot be obtained by scaling of features.
>
> Finally, based on current advances in FSL and causal inference, we cannot infer the exact causal or reasoning path from our action (i.e., removing the projection to the task centroid) to the final classification. However, we do believe that our findings on the correlation between the distance to the task centroid  and the accuracy distribution (as illustrated in Fig. 1b) may shed a light on the importance of conducting interpretative feature analysis in FSL, in contrast with the fact that most deep models in FSL literature have only a rough motivation without any verification except for the accuracy. We expect that  the FSL community could start from our feature analysis and discover deeper properties of image features (including the derivation of the reasoning path from the task centroid to final generalization bound) in few-shot learning.

---

> > ### Author Response · Authors · 2022-08-02
> > **Response to Reviewer 5bg4(2/2)**
> >
> > ### **Q3: I am not convinced that the change of origin is the source of bias. Somehow I feel that the algorithm tries to use the base data to construct a better latent space. A natural question to ask is, can we consider the PCA space from the base samples as a way of denoising the decision space? That is, aside from the task-center, what will happen if the PCA space is also taken into account. Can you kindly comment on this?**
> >
> > **A3**: To help the reviewer better understand the difference between our proposed method to PCA,
> > we follow the suggestion to use PCA on the base dataset to learn a linear projection that projects original features into an n-dimensional subspace. This linear projection is then used to transform all features in the novel set. In addition, we also try to use PCA on the whole novel set to learn a  linear projection. The 1-shot performance is shown below.
> >
> > | dimension $n$| PCA on base    | PCA on novel   |
> > |-|-|--|
> > | 640(no pca)   |92.86$\pm$0.38|92.86$\pm$0.38|
> > | 400 | 90.65$\pm$0.35 | 93.52$\pm$0.28|
> > | 100  | 89.71$\pm$0.38| 92.19$\pm$0.32|
> > | 10 | 84.08$\pm$0.52| 89.86$\pm$0.45|
> > | 3   | 79.68$\pm$0.72 | 86.53 $\pm$0.71|
> >
> > The performance decreases when PCA is conducted on base data, while the performance first increases and then decreases when PCA is conducted on novel data. Since PCA remains the most informative direction of dataset, we can infer that some non-informative directions of base dataset should be important for discriminating novel dataset, while non-informative directions of novel dataset are indeed not important or even harmful for the novel set itself. In other words, the noise in base dataset is not the real noise in novel dataset, thus using PCA on base dataset to denoise the feature space is inappropriate. **On the contrary, although our method TCPR uses base dataset to estimate the non-discriminative dimension (the direction along the task centroid), we seek for base features that are most similar to the novel task, thus we are still estimating the dimension in the novel dataset.** TCPR can lead to a 3\% improvement(from 92.86\% to 95.79\%). Note that the noisy dimension found by TCPR (task centroid) may be very different from the noise found by PCA on the whole novel set, and it is future work to investigate the relationship. While PCA on the whole novel set could improve performance, it uses the global dataset knowledge and thus cannot be applied to few-shot learning.
> >
> > ### **Q4: the problem studied is solely classification.**
> >
> > **A4**: We respectively disagree with the reviewer that this is a weakness. The literature of the classical few-shot learning methods (e.g., MAML, MatchingNet and ProtoNet) and literature of current state-of-the-arts (e.g., DC and Inv-Equ) studied the problem of classification solely. For fair comparison with those methods, we conduct the same classification task. Although it would be interesting to study other learning tasks, it is beyond the scope of this paper.
> >
> > ### **Q5: Can the method be used to address dense prediction FSL? if yes, how?**
> >
> > **A5**: Although dense prediction FSL is beyond the scope of this paper, we agree it would be interesting to apply our method to dense prediction FSL tasks like object detection/segmentation, and we plan to explore it in the near future.
> > Since dense prediction tasks care more about positional information and always use the spatial feature instead of the flattened one, one way is to implement our TCPR method directly on each individual spatial feature in the spatial feature map. The support centroid now turns to be the mean of all individual spatial features in the support set. Looking for similar base features also changes to looking for similar individual spatial features in the base datasets, which means it is possible that only parts of individual spatial features in one image are chosen. Then the dimension along the approximated task centroid will be removed from every individual spatial features in the novel set. Since we only transform the feature and do not change anything else, this extended method can be applied to any existing dense prediction FSL methods. We will conduct preliminary experiments in the near future.

---

> ### Author Response · Authors · 2022-08-08
> **To Reviewer 5bg4**
>
> Dear Reviewer 5bg4,
>
> We thank you for your review and appreciate your time reviewing our paper.
>
> The end of the rebuttal phase is approaching. We would be grateful if we could hear your feedback regarding our answers to the reviews. We are happy to address any remaining points during the remaining period.
>
> Thanks in advance,
>
> Paper4931  authors

---

### Official Review · Reviewer_i4sq · 2022-07-11

**Rating:** 6
**Confidence:** 4
**Soundness:** 3 good
**Presentation:** 3 good
**Contribution:** 3 good

**Summary:**

This paper proposes a novel few-shot learning method of task centroid projection removing (TCPR). The proposed TCPR aims to resolve one of the limitations of few-shot learning, called the sampling bias problem. By adjusting the classification boundary via moving the data centroids by removing some components of features, TCPR can improve the few-shot classification accuracies for various datasets, feature extractors, and base few-shot classification algorithms.

-----------------------------------------------------------------------------------------------------------------------------------------------------------------------------------------
After Rebuttal: I appreciate the authors for kindly addressing my concerns and questions. Most of my concerns are resolved. As a result, I keep my original score.

**Questions:**

1. Related to weakness 2, can we utilize the base class centroids instead of the base class data? Since there exist many data for the base classes, we can easily compute the true class centroids for the base classes. Isn’t it more realistic to store the base class centroids, not the entire base class data? How does this affect performance?
2. The process of removing the projection to the task centroid seems to make the features orthogonal to the task centroid and remove the effect of the dimension of task-centroid. Is there any other way to remove the dimension of task-centroid? For example, [1] utilizes the linear nulling of the vector to remove the effect of specific dimensions.

[1] Yoon et al. TapNet: Neural Network Augmented with Task-Adaptive Projection for Few-Shot Learning, ICML 2019.


**Limitations:**

1. Sampling process for computing task centroid may be computationally inefficient.
2. To obtain the gain of TCPR, we need to store some extra base data, which can be unreal condition.

**Strengths And Weaknesses:**

Strengths

1. Idea of approximating task centroid with the help of base class data is interesting and powerful.
2. The concept of utilizing the task centroid to resolve sampling bias problem is intuitive and powerful, and it can be easiliy applied to various FSL techniques. Experimental results show that the proposed TCPR is able to improve the performances of various few-shot learning methods.
3. Analysis results on sampling bias problem would be helpful for further research on sampling bias problem.

Weakness

1. It seems the process of sampling the most related samples from the base class samples seems computationally inefficient. Since there are many base class data and N_base is large, it seems that computing cosine similarity and choosing top-K samples from base data require many computations.
2. The task centroid has to be computed anew for the given task, and we need to keep the entire base data, not just the trained model. For me, it seems that keeping all the base class samples for the approximation of task centroid seems somewhat unreal.
3. I am not sure whether the comparing TCPR-extended few-shot learning with the base few-shot learning model. In Figure 7, it seems that TCPR requires the large k to obtain meaningful gain, which means TCPR utilizes many extra data, although the extra data come from the base classes.

---

> ### Author Response · Authors · 2022-08-02
> **Response to Reviewer i4sq**
>
> We appreciate the reviewer for positive comments. Below please find our responses to the questions.
>
>
> ### **Q1:  it seems that computing cosine similarity and choosing top-K samples from base data require many computations.**
>
> **A1**: Indeed, the time cost to get top $k$ base samples is not expensive. To demonstrate this, we provide the latency (ms) of one test-time 5-way 1-shot task on miniImageNet shown below. The latency is averaged over $2000$ tasks. The increase in latency is relatively small ($11$ms) compared with the original time of training the classifier ($298$ms). We will add this result into the final version of the paper.
>
>
> | Selection Ratio      | Latency | mini-test       | CUB             | Fungi           | Omini            | Sign            | QDraw           | Flower          | DTD             |
> |----------------------|---------|-----------------|-----------------|-----------------|------------------|-----------------|-----------------|-----------------|-----------------|
> | baseline             | 298     | $64.63$         | $47.75$         | $42.36$         | $77.28 $         | $53.50$         | $51.60$         | $70.33$         | $50.47$
> |TCPR           | + 11    | $68.06$ | $51.87$ | $44.38$ | $78.51 $ | $54.83$ | $54.62$ | $72.55$ | $52.50$ |
> | TCPR(ratio=$0.01$)    | +4      | $67.79$           | $51.55$           | $44.29$           | $78.18$            | $54.03$           | $53.62$           | $71.82$           | $52.69$           |
> | TCPR(base proto) | +4      | $67.67 $          | $51.27$           | $44.15$           | $77.83$            | $53.76$           | $53.16  $         | $71.70$           | $52.50$           |
>
> ### **Q2: The required storage.**
>
> **A2**: Since our method works on the feature level, we need to store only the features of the base data (which are 640 dimensions in our experiments) instead of the entire training data. For example, all training features of miniImagenet require only 98M storage space, which is acceptable compared to the overall overhead.
> Moreover, it is possible to access only a small subset of the training features. We randomly sample a small ratio(e.g., $0.01$) of features in each base class with $0.01$ storage space(0.98M), the improvements over baseline methods are still significant, as seen in the table above. Especially, when keeping one proto-type for each base class like DC, the proposed method has a small drop over using the whole training features.
>
> ### **Q3: Fair comparison.**
>
> **A3**:The aim of few-shot learning is to utilize prior knowledge (from the base dataset) to quickly adapt to novel tasks, and it does not matter what the form  of the knowledge is. It is just recent years that neural networks serve as a popular abstract container of the knowledge, but this should not be regarded as the golden rule. On the contrary, humans can memorize typical instances of objects and quickly retrieve similar ones when observing an unknown object to help classification. Our method is to some extent similar to such form of memorization and retrieval. Indeed, hallucination-based few-shot learning methods always utilize the original base dataset in addition to the learned network to help augment samples at test time [1-4].
>
> [1] Low-shot Visual Recognition by Shrinking and Hallucinating Features. ICCV 2017.
>
> [2] Image Deformation Meta-Networks for One-Shot Learning. CVPR 2019.
>
> [3] Delta-encoder: an effective sample synthesis method for few-shot object recognition. NeurIPS 2018.
>
> [4] Free Lunch for Few-shot Learning: Distribution Calibration. ICLR 2021.
>
> ### **Q4: TapNet and other ways to remove the bad effect of task centroid**
>
> **A4**: Note that in TapNet, the dimensions removed by linear nulling are the directions of  $\epsilon_k$, i.e., the direction of the line connecting each class centroid with corresponding modified reference vectors. Since these modified reference vectors are meta-trained on base dataset and fixed at test time, there's no guarantee that one of the removed dimensions estimated from support classes aligns with the task centroid. However, if we set the reference vector to 0 (although deviating from the original paper that each reference vector represents a class), at this time, the removed dimensions are exactly all directions along classes centroids. Since the task centroid is a linear combination of class centroids, removing these dimensions also removes the dimension along task centroid. This perspective could be very interesting, and we will explore more in the near future. Thanks for your valuable suggestion! Since estimating the task centroid by similar base features has some estimation errors, we believe there are other methods as well, and we leave the exploration as future work.

---

### Official Review · Reviewer_ZuAk · 2022-07-11

**Rating:** 5
**Confidence:** 5
**Soundness:** 2 fair
**Presentation:** 3 good
**Contribution:** 3 good

**Summary:**

This paper studies the sampling bias issue in few-shot learning and identifies its connection to task centroids. To mitigate the sampling bias, this paper proposes to perform Task Centroid Projection Removing (TCPR) before training the classifier for the test task. When performing TCPR, it approximates the task centroid by using top K data samples in base classes. Experimental results show the proposed method improves the classification accuracy for different pre-trained feature extractors.

**Questions:**

1. The concern about the large value of k. Could the authors provide experiment results about the running time cost of obtaining top k base samples and training the classifier?
2. The toy case in Sec 3.3. TCPR cannot be applied in this case. I think it would be better to project the data into a 3-d hypersphere and perform TCPR to see how TCPR helps to improve the performance. I think TCPR would work for some cases and fail for other cases.
3. TCPR is somehow similar to DC[43] as both methods try to improve the performance by using the information of base samples. I think it would be better to provide more discussion about DC in related works and provide more experiment results of DC on the same pre-trained feature extractors as TCPR, e.g. Baseline++, Inv-Equ, S2M2, S2M2*. Besides, is it possible to retrieve top k base classes prototypes (like DC) instead of top k base samples, so that not all base class samples are required?
4. About Experiment in Table 3: It would be desirable to provide the distance between the approximation of the different task centroids and the real task centroid. In addition, the results of the 5-shot experiments should be provided in Table 3 for a better understanding of the improvement of c_task on the 5-shot experiments.

**Limitations:**

The paper discusses the limitation about selection of k.

**Strengths And Weaknesses:**

Originality:
The studies about the connection between sampling bias and task centroid and the proposed TCPR method are novel.

Quality:
The proposed TCPR is technically sound.
The paper tries to support the claim about the connection between sampling bias and task centroid through toy cases and experiments. But the toy case in Sec 3.3 is not a good toy case, because it does not match with the cases where TCPR is applicable.
Another issue is computation cost. The value of k is very large. The time cost to get top k base samples may be high. Moreover, this method requires access to all base samples, which may be not practical.


Significance:
The sampling bias is an important issue in few-shot learning. This paper studies the connection between sampling bias and task centroid and proposes TCPR to mitigate it. Applying TCPR on various pre-trained feature extractors, significant performance improvements are achieved.

Clarity:
The paper is not difficult to understand. It clearly presents the motivation and the method and provides sufficient technique and experiment details. But the writing can be further improved.

----
After rebuttal:
The response from the authors addresses my questions. I still hold some minor concerns about the failure of TCPR on the toy case, as the authors did not provide a revised version of the paper, and also about the large value of k.
I keep my score weak accept.

---

> ### Author Response · Authors · 2022-08-02
> **Response to Reviewer ZuAk(1/2)**
>
> We appreciate the reviewer for insightful comments. Below please find our responses to the questions.
>
> ### **Q1:The time cost to get top k base samples may be high.**
>
> **A1:** Indeed, the time cost to get top $k$ base samples is not expensive. To demonstrate this, we provide the latency (ms) of one test-time 5-way 1-shot task on miniImageNet shown below. The latency is averaged over $2000$ tasks. The increase in latency is relatively small ($11$ms) compared with the original time of training the classifier ($298$ms). We will add this result into the final version of the paper.
>
> | Selection Ratio      | Latency | mini-test       | CUB             | Fungi           | Omini            | Sign            | QDraw           | Flower          | DTD             |
> |----------------------|---------|-----------------|-----------------|-----------------|------------------|-----------------|-----------------|-----------------|-----------------|
> | baseline             | 298     | $64.63$         | $47.75$         | $42.36$         | $77.28 $         | $53.50$         | $51.60$         | $70.33$         | $50.47$
> |TCPR           | + 11    | $68.06$ | $51.87$ | $44.38$ | $78.51 $ | $54.83$ | $54.62$ | $72.55$ | $52.50$ |
> | TCPR(ratio=$0.01$)    | +4      | $67.79$           | $51.55$           | $44.29$           | $78.18$            | $54.03$           | $53.62$           | $71.82$           | $52.69$           |
> | TCPR(base proto) | +4      | $67.67 $          | $51.27$           | $44.15$           | $77.83$            | $53.76$           | $53.16  $         | $71.70$           | $52.50$           |
>
>
> ### **Q2:Moreover, this method requires access to all base samples, which may be not practical.**
>
> **A2**: Since our method works on the feature level, we need to store only the features of the base data (which are 640 dimensions in our experiments) instead of the entire training data. For example, all training features of miniImagenet require only 98M storage space, which is acceptable compared to the overall overhead.
> Moreover, it is possible to access only a small subset of the training features. We randomly sample a small ratio(e.g., $0.01$) of features in each base class with $0.01$ storage space(0.98M), the improvements over baseline methods are still significant, as seen in the table above. Especially, when keeping one proto-type for each base class like DC, the proposed method has a small drop over using the whole training features.
>
> ### **Q3:About projecting data into a 3-d hypersphere.**
> **A3**: Following the suggestion, we randomly choose two classes and project all data into a 3-dimensional subspace using PCA, and then use TCPR to remove the projection along the task centroid in the subspace. The visualization results are shown in https://anonymous.4open.science/r/FSL-TCBR-3208/README.md  ( if the link fails, please refer to the code in the Supplementary), which demonstrates that the features of different classes are pushed away by TCPR, especially for the features in the vicinity of the centroid of data.  The simulation is shown in Figure 2(b) is the paper.
>
> To further demonstrate the effectiveness of TCPR in $3$-d settings, we design a new toy case of $3$-d with Gaussian distribution where TCPR is applicable. TCPR results in a 4.2\% accuracy improvement (from 65.99\% to 69.17\%). The simulation code can be found in https://anonymous.4open.science/r/FSL-TCBR-3208/gaussian_simulation.py.  ( if the link fails, please refer to the code in the Supplementary)
>
> We understand that TCPR cannot be applied to our toy case because the task centroid coincides with the origin. We have done experiments and found that in realistic datasets, the task centroids are generally far from the origin. We agree that there exist some artificial special cases where our method may fail. But we also believe these cases do not exist in realistic data due to some intrinsic nature of images and inductive bias brought by neural network architecture and optimizations. Exploring this would be interesting, and if timely, we will involve more analysis in the final version of our paper.

---

> > ### Author Response · Authors · 2022-08-02
> > **Response to Reviewer ZuAk(2/2)**
> >
> >
> > ### **Q4:The Comparison with DC.**
> >
> > **A4**:Thanks for the suggestion. We compare the performance of DC with our method on Meta-dataset shown below. The backbone is trained on miniImageNet using S2M2. Note that DC degrades performance drastically in cross-domain setting, i.e., when not evaluated on mini-test. This phenomenon originates from the motivation of DC: using base class statistics to generate fake features in order to augment the support set. When the domain gap exists, there are always no base classes having statistics similar to the given novel class. In fact, it is likely that the discriminative pattern of the novel task is not the emphasis in any of base classes. Instead, a new cluster of base features that emphasizes discriminative patterns would be more appropriate for estimating the class statistics of the novel set. thus DC would fail to improve performance. On the contrary, our method does not consider the class information in the base dataset, and thus can flexibly seek similar base features to form a cluster for estimating the task centroid. We observe similar phenomenon when using Baseline++ or Inv-Equ as backbone, and we will include all results in the final version of our paper.
> >
> > |          | mini-test       | CUB             | Fungi           | Omini            | Sign            | QDraw           | Flower          | DTD             |
> > |----------|-----------------|-----------------|-----------------|------------------|-----------------|-----------------|-----------------|-----------------|
> > | baseline | 65.53           | 48.75           | 43.11           | 75.62            | 53.41           | 50.60           | 71.05           | 51.10           |
> > | + DC     | 68.46           | 48.06           | 42.39           | 74.66            | 52.58           | 47.93           | 66.50           | 49.40           |
> > | + TCPR | 68.06 | 51.87 | 44.38 | 78.51 | 54.83 | 54.62 | 72.55 | 52.50 |
> >
> > ### **Q5:Experiment in Table3.**
> >
> > **A5**: We provide the results of 5-way 5-shot tasks as below. Similar with 1-shot task,  TCPR gets consistent improvement on Meta-dataset. The cosine distances between our approximation and the real centroid is also as below.
> >
> >
> > | Test Set   | mini-test | CUB   | Fungi | Omini | Sign  | QDraw | Flower | DTD   |
> > |------------|-----------|-------|-------|-------|-------|-------|--------|-------|
> > | baseline   | 83.54     | 70.24 | 62.38 | 93.12 | 74.54 | 72.87 | 89.70  | 71.67 |
> > | TCPR + $\bar{x}^n_{\mathcal C_{novel}} $       | 84.54     | 71.51 | 63.46 | 93.71 | 75.47 | 74.32 | 90.31  | 72.76 |
> > | TCPR + $\bar{x}^n_{\mathcal S} $     | 84.20     | 70.95 | 62.88 | 92.81 | 74.78 | 73.56 | 89.96  | 72.26 |
> > | TCPR + $\mathbf c_{task}$       | 84.51     | 71.24 | 63.12 | 93.52 | 74.99 | 73.78 | 90.11  | 72.21 |
> > | SimpleShot | 84.47     | 70.62 | 62.64 | 93.22 | 74.81 | 73.18 | 89.83  | 71.88 |
> > | ZN         | 83.99     | 70.88 | 63.11 | 93.30 | 75.01 | 73.54 | 89.89  | 71.81 |
> > | distance         | 0.96     | 0.95 | 0.92 | 0.91 | 0.91 | 0.93 | 0.94  | 0.91 |

---

### Official Review · Reviewer_mLQK · 2022-07-11

**Rating:** 4
**Confidence:** 4
**Soundness:** 3 good
**Presentation:** 3 good
**Contribution:** 3 good

**Summary:**

This paper claims to be the first to investigate bias of the supports that causes low accuracy in few-shot learning. Specifically, when the supports are too close to the centroid of a task, the quality of the decision boundaries would be unstable and hence a lower accuracy. They propose a framework to remove such bias by firstly estimating the centroid with base and support features from a pre-trained feature extractor and removing the components that lie in the same direction as the centroid, pushing away the novel features, and thus mitigating the bias. With the experiments on several benchmarks, they showed the effectiveness of the method.

**Questions:**

1.	The assumption is novel features tend to mingle with base features, and one could find K nearest of them for task centroid estimation. Any guarantees on this? What if novel features lie in a sparse region where the closest base features are far? Would it cause instability for centroid estimation? More exploration and evidence on this would be greatly appreciated and would bolster the work.

2.	The simulation experiments are in 2-d, difference in high dimensional space?

3.	I would imagine in some cases, removing the projection to the centroid could potentially cause performance degradation? For instance, what if supports from two classes lie in the upper sphere while supports for the other two lie in the other sphere? Then, in this case, initially we could use the direction along the centroid for differentiation, and yet after projection, we might not be able to?


**Limitations:**

No discussion on limitations.

**Strengths And Weaknesses:**

Strengths

1.	The idea is intuitive and easy to understand and follow, which they also show the effectiveness by applying the method on different pre-trained few-shot models.
2.	They showed SoTA performance on different benchmarks.
3.	A set of ablation studies are performed on the design choices.

Weaknesses

1.	No analysis on failure cases, if any. Discussion on limitations is limited.

---

> ### Author Response · Authors · 2022-08-02
> **Response to Reviewer mLQK (1/2)**
>
> We thank the reviewer for the exploratory feedback on our work. Although the reviewer's concerns about the feature distribution are interesting, we find out those extreme situations are not existing in real data in FSL.  Please find the responses below.
>
> ### **Q1: The assumption is novel features tend to mingle with base features, and one could find K nearest of them for task centroid estimation. Any guarantees on this? What if novel features lie in a sparse region where the closest base features are far? Would it cause instability for centroid estimation? More exploration and evidence on this would be greatly appreciated and bolster the work.**
>
> **A1**: That is a very good question.   As the features of both the base data and novel data are pre-trained through the same backbone network, it is highly unlikely that novel features lie in a sparse region where the closest base features are far. To verify this point,
> we calculate the feature similarity between the base data and novel data in the few-shot learning setting. To make it more challenging, we choose the Meta-dataset where there is a very large domain gap where the novel data domains (e.g., Quickdraw and Omniglot) are prominently different from the base data (i.e., miniImageNet). In detail,
> we define $\lambda$ as the cosine similarity between the centroid of the support data and the base data. When we use $\lambda>0.6$ as the condition to select the most similar neighborhoods, we see that the neighborhood size varies from 290 (when Omini is set as the novel set) to 8168 (when miniImage-test is set as the novel set). In these settings, we observe stable improvement over the baseline method (i.e., S2M2).
> This suggests that we can find k-nearest neighbors from the base data for the centroid estimation in current FSL evaluation settings. There are two potential explanations. Firstly, realistic images (even from different domains) tend to share some common underlying characteristics (e.g., low-level cues like texture or color which are verified to be shared across domains [1]). Secondly, recent FSL methods generally pretrain a  backbone network on base data via contrastive learning to improve the generalization ability, while representations learned by contrastive learning uniformly locate on the hypersphere[2]. Thus the projection of novel data to the base data can easily find neighborhoods in dense regions.
>
> |                         | mini-test         | CoCo               | CUB              | Fungi            | Omini            | Sign            | QDraw           |
> |-------------------------|-------------------|-------------------|------------------|------------------|------------------|-----------------|-----------------|
> | baseline                | $64.63$           | $63.06$           | $47.75$          | $42.36$          | $77.28 $         | $53.50$         | $51.60$         |
> | TCPR               | $68.06_{k=10000}$ | $64.32_{k=5000}$ | $51.87_{k=5000}$ | $44.38_{k=5000}$ | $78.51_{k=100} $ | $54.83_{k=100}$ | $54.62_{k=100}$ |
> | TCPR($\lambda>0.6$) | $67.72_{k=8168}$  | $63.81_{k=6032}$   | $51.57_{k=6437}$ | $44.27_{k=4092}$ | $78.06_{k=290} $ | $54.22_{k=890}$ | $53.56_{k=367}$ |
>
>
> Moreover, our automatic way of choosing the value of $k$ works for all datasets, with accuracy close to the best one found by grid search.
>
> [1] Zhao N, et al. what makes instance discrimination good for transfer learning? ICLR 2021.
>
> [2] Wang T, Isola P. Understanding contrastive representation learning through alignment and uniformity on the hypersphere[C]//International Conference on Machine Learning. PMLR, 2020: 9929-9939. ICML 2019.
>
>
> ### **Q2:The simulation experiments in high dimensional space.**
>
> **A2**:We offer the simulation experiments in higher dimensional space(i.e., 3/10/50-dimension). Please refer to https://anonymous.4open.science/r/FSL-TCBR-3208/README.md for the simulation ( if the link fails, please refer to the code in the Supplementary). The simulation code is in ``gaussian_acc.py''. The similar phenomenon can be observed.

---

> > ### Author Response · Authors · 2022-08-02
> > **Response to Reviewer mLQK (2/2)**
> >
> >
> > ### **Q3:What if supports from two classes lie in the upper sphere while supports for the other two lie in the other sphere? Then, in this case, initially we could use the direction along the centroid for differentiation, and yet after projection, we might not be able to?**
> >
> > **A3** :If we understand the question correctly, the reviewer thinks it is possible that the line connecting two classes is parallel to the direction of the task centroid, thus removing the direction of task centroid may lead to loss of discriminative power. However, we found that it is impossible in real-world high-dimensional data. To see this, we conduct an additional experiment. We randomly sample 5 classes in the test-time dataset (i.e., 5-way task). Then we calculate the cosine similarity between the task centroid and those lines connecting all possible pairs of classes. Thus for such a task we calculate 10 similarities (i.e., 10 possible combinations of classes). For each test-time dataset, we sample 2000 such 5-way tasks and report the average similarity as well as the maximum similarity among all $20000$ (i.e. $2000
> > \times10$ calculated similarities. The results are shown in the table below.
> >
> > |      | mini  | Cub   | Dtd   | Fungi | Omini | Sign  | Qdraw | Flowers |
> > | ---- | ----- | ----- | ----- | ----- | ----- | ----- | ----- | ------- |
> > | Avg  | 0.039 | 0.023 | 0.029 | 0.020 | 0.012 | 0.017 | 0.006 | 0.021   |
> > | Max  | 0.191 | 0.197 | 0.162 | 0.205 | 0.087 | 0.108 | 0.065 | 0.141   |
> >
> >
> > As seen from the table, in each dataset, even the maximum similarity among 20000 possibilities is far less than 1, and the average similarity is extremely low. This indicates that the line connecting two classes is very likely to be orthogonal to the direction of task centroid. We conjecture that this is due to the nature of the high dimension of feature space: in a high-dimensional space, two randomly sampled vectors are very likely to be orthogonal to each other.
> >
> > ### **Q4**: The failure cases:
> >
> > **A4**: Although we could imagine some failure cases in extreme cases, we have not yet observed them in realistic data in the $640$ dimensional space of novel classes.  Please see our detailed explanations in Q3.  Besides, Figure 1(b) is conducted on the real novel data, showing the bias problem caused by task centroid and the effectiveness of the proposed TCPR. Similar phenomena can be observed in other novel data.
> >
> > ### **Q5**: No discussion on limitations.
> > **A5**:  We have discussed the limitation of selecting the number of $k$ neighbors in the Future Work section. Besides, the answer in Q1 can be seen as a simple way to select the $k$ adaptively. We will explore more powerful solutions in future.

---

> > > ### Comment · Reviewer_mLQK · 2022-08-09
> > > **Response to authors’ rebuttal**
> > >
> > > I thank the authors for the additional and insightful experiments and visualization. The authors have addressed most of my concerns, especially showing that in practice, it’s unlikely that the features from novel classes would be far away from the base.

---

> > > > ### Author Response · Authors · 2022-08-10
> > > > **Thank you for the reply!**
> > > >
> > > > Dear reviewer, thanks again for your valued advice! Since most of your concerns have been clarified, could you please reconsider the score?

---

> ### Author Response · Authors · 2022-08-08
> **To Reviewer mLQK**
>
> Dear Reviewer mLQK,
>
> We thank you for your review and appreciate your time reviewing our paper.
>
> The end of the rebuttal phase is approaching. We would be grateful if we could hear your feedback regarding our answers to the reviews. We are happy to address any remaining points during the remaining period.
>
> Thanks in advance,
> Paper4931  authors

---

### Official Review · Reviewer_XMzd · 2022-07-11

**Rating:** 8
**Confidence:** 4
**Soundness:** 4 excellent
**Presentation:** 3 good
**Contribution:** 4 excellent

**Summary:**

This paper presents a new technique, Task Centroid Projection Removing (TCPR) for few-shot classification. The paper highlights the phenomenon of the distance of samples from a task centroid affecting task performance. The paper then builds on this insight to create the TCPR technique.

**Questions:**

Why is cosine distance used for all similarity measurements? Could any distance metric be used?

On line 233, what are ‘cosine classifiers’? Could you give a more thorough explanation of how the final classification was done for the query set.

**Limitations:**

The authors have addressed the limitations of their work.

**Strengths And Weaknesses:**

___Strengths___: The paper is attacking a significant problem in machine learning and is presenting a novel technique to address that significant problem. Learning with less labels, which includes few-shot learning, is one of the greatest problems hindering machine learning. So, this paper is directly addressing a very significant problem. The technique proposed as well as the insight about support samples’ positions in the embedded space is rather novel and straightforward. The idea of pre-training bias affecting subsequent few-shot classification is novel and also reflects some of the known wisdom regarding domain adaptation in machine learning. The use of base set in constructing the task centroid is a clever and straightforward means of controlling the variability that can arise in sampling for few-shot classification.

The paper also does a thorough empirical evaluation, including an ablation study with different models and hyperparameters. The thoroughness of this study combined with the released code greatly inspires the reader to try the proposed technique on their own work.

___Weaknesses___: The paper has a few weaknesses. First, there are some minor clarity issues. The placement of figures is wonky in that Table 3 comes before Table 2. And, while this is an acknowledged limitation, the setting of $k$ for the number of nearest neighbors really needs more guidelines for practical use.

---

> ### Author Response · Authors · 2022-08-02
> **Response to Reviewer XMzd**
>
> We thank the reviewer for the positive review and insightful comments. We respond to your questions below.
>
> ### **Q1:the setting of $k$ for the number of nearest neighbors really needs more guidelines for practical use.**
>
> **A1**: That is a good point and is a question we want to address in future work. A simple and intuitive solution could be using the cosine similarity (denoted by $\lambda$) between the centroid of support data and base data to automatically select the top $k$ neighbors. We use $\lambda>0.6$ as the condition to select the most similar neighborhoods. We list the adaptively selected values of $k$ (averaged over $2000$ tasks) and the corresponding performance of TCPR. These preliminary results demonstrate the effectiveness of the adaptive setting, and we will include them in the final version of the paper and explore more powerful solutions in future work.
>
> |                         | mini-test         | CoCo               | CUB              | Fungi            | Omini            | Sign            | QDraw           |
> |-------------------------|-------------------|-------------------|------------------|------------------|------------------|-----------------|-----------------|
> | baseline                | $64.63$           | $63.06$           | $47.75$          | $42.36$          | $77.28 $         | $53.50$         | $51.60$         |
> | TCPR               | $68.06_{k=10000}$ | $64.32_{k=5000}$ | $51.87_{k=5000}$ | $44.38_{k=5000}$ | $78.51_{k=100} $ | $54.83_{k=100}$ | $54.62_{k=100}$ |
> | TCPR($\lambda>0.6$) | $67.72_{k=8168}$  | $63.81_{k=6032}$   | $51.57_{k=6437}$ | $44.27_{k=4092}$ | $78.06_{k=290} $ | $54.22_{k=890}$ | $53.56_{k=367}$ |
>
>  ### **Q2: Why is cosine distance used for all similarity measurements? Could any distance metric be used?**
>
> **A2**: Our method is agnostic to the metric being used. Below we show the results when we use the euclidean distance:
>
> |          | S2M2  | Inv-Equ | Baseline++ |
> |----------|-------|---------|------------|
> | baseline | 61.78 | 62.34   | 56.34      |
> | +TCPR    | 62.72 | 63.20   | 57.41      |
>
>
> ### **Q3:The cosine classifier**
>
> **A3**: In the cosine classifier, the prediction is based on the cosine distance between the input feature and the learned weight vectors representing each class, which is firstly introduced in Baseline++[1].
>
> The cosine classifier is learned on the support data with a normalized weight vector for each novel class. After learning, given a query/test example, we calculate its cosine similarity with each normalized weight vector. The query example is then classified as the class with the highest similarity score. Please refer to [1] and our code for the details.
>
>
> [1] Chen W Y, Liu Y C, Kira Z, et al. A closer look at few-shot classification[J]. ICLR 2019.

---

### Official Review · Reviewer_HDBL · 2022-07-12

**Rating:** 6
**Confidence:** 3
**Soundness:** 3 good
**Presentation:** 2 fair
**Contribution:** 3 good

**Summary:**

The paper conducts experiments which show that classification boundary in a few-shot setting is very sensitive to the position of support samples (few examples seen during evaluation) if they are in the vicinity of the data centroid (referred to as task centroid). Hence, a feature transformation named Task Centroid Projection Removing (TCPR) is proposed to remove the component of features along the direction of approximated task centroid. Experimental evaluation is conducted on several datasets.

**Questions:**

Is the "Removing the projection to the task centroid" similar to VLAD/Fisher Vector encodings which have previously been used in image classification? Is it possible to distinguish with these methods?

[1] https://openaccess.thecvf.com/content_cvpr_2015/papers/Xu_A_Discriminative_CNN_2015_CVPR_paper.pdf This was used for few shot event recognition.

sentence correction:

99: DCM [36] using support data to improve the evaluation performance calibrates the distribution of novel samples to approach zero centered mean and unit standard deviation .

using support data to improve the evaluation performance calibrates the distribution  -> uses support data to improve the performance by calibrating the distribution

101: our method has different motivation to reduce the classification bias when the support samples in the vicinity of local centroid.

Not sure what this means

**Ethics Review Area:**

["I don’t know"]

**Limitations:**

yes, they are addressed

**Strengths And Weaknesses:**

Strengths:
The paper motivates the problem well with the “Lion” and “Dalmatian” example which highlights the problem with existing techniques. Later on, the paper addresses this problem with a simple solution by picking nearest samples from the base class and then picking a vector which points in the direction along the k nearest points in the base class, along a direction away from the class mean. Extensive experiments show the effectiveness of the method.

Weakness:
Some of the terminology being used in the paper is a bit non-intuitive to understand and this being coupled with incorrect sentences makes the paper hard to read.

---

> ### Author Response · Authors · 2022-08-02
> **Response to Reviewer  HDBL**
>
> We thank the reviewer for providing insightful comments. Our response to your questions is as follows:
>
> ### **Q1:Is the "Removing the projection to the task centroid" similar to VLAD/Fisher Vector encodings previously used in image classification? Is it possible to distinguish between these methods?**
>
> **A1**: VLAD/Fisher encodings are extensions of the bag-of-word method aiming at obtaining meaningful image features. These methods assume that each feature belongs to one of the feature clusters (or Gaussian components), and use the feature statistics w.r.t the corresponding cluster to represent an image. Thus the motivation is not about removing the sampling bias.
>
> If we set the number of clusters in VLAD to one (which would go against VLAD's motivation), then VLAD encodings subtract each feature by the dataset centroid and thus look similar to our method. However, subtracting each feature only by the dataset centroid will not change the dataset pattern, not to say the distance of any feature to the dataset centroid. Instead, our method subtracts each feature by the projection onto the dataset centroid, which removes the dimension along the dataset centroid and obtains a subspace of N-1 dimensional. In this way, the distance from features to the dataset centroid is adjusted differently, and features close to the task centroid are pushed further away from the centroid.
>
> ### **Q2:Some non-intuitive terminology and incorrect sentences.**
>
> **A2**: We have improved the writing in the revised paper.

---

### Meta-Review · Area_Chair_5cmc · 2022-08-26

**Recommendation:** Accept
**Confidence:** Certain

**Metareview:**

The paper investigates the bias of support points in few shot learning when they are too close to the task centroid. This can cause a drop in accuracy. It proposes a method to mitigate this by projecting out the task centroid direction. Initial reviews were split on the paper with three reviewers having a positive opinion while the other three had concerns. However the author response has addressed many of the reviewer concerns (also acknowledged by one of the reviewers). The paper makes a solid contribution to the few shot learning problem and will be a good addition to the program.

**Award:**

No

---

### Decision · Program_Chairs · 2022-09-14

Accept